# Wrinkle Reduction Using Tetrapeptide-68 Contained in an O/W Formulation: A Randomized Double-Blind Placebo-Controlled Study

**DOI:** 10.3390/pharmaceutics16080987

**Published:** 2024-07-25

**Authors:** Sung-Gyu Lee, Sang-Moon Kang, Hyun Kang

**Affiliations:** 1Department of Medical Laboratory Science, College of Health Science, Dankook University, Cheonan-si 31116, Republic of Korea; sung-gyu@dankook.ac.kr; 2R&D Center, A&PEP Inc., Cheongju-si 28101, Republic of Korea; smkang@anpep.com

**Keywords:** peptide, anti-wrinkle, Tetrapeptide-68 cream, anti-aging, clinical trial

## Abstract

Peptides, composed of 2–50 amino acids, have gained attention in anti-aging treatments due to their high safety, low irritation, and cost-effective production. This study aimed to evaluate the anti-wrinkle efficacy of Tetrapeptide-68, derived from the skin structural protein Loricrin, on periorbital wrinkles in women aged 30–65 years. A 12-week, double-blind, randomized controlled trial was conducted with 25 participants who applied the Tetrapeptide-68 (100 ppm) O/W formulation around the eyes. Skin physiological parameters were assessed at baseline, 4, 8, and 12 weeks. Participants also completed efficacy and usability questionnaires. Significant improvements in wrinkle reduction were observed with Tetrapeptide-68 cream treatment, as measured by various skin roughness parameters and 3D imaging analysis. Participants reported positive changes in skin texture and moisture levels, with no adverse reactions noted. Tetrapeptide-68 cream demonstrates promising anti-wrinkle effects, highlighting its potential as an effective ingredient in anti-aging skincare formulations. Further studies are recommended to explore its long-term benefits and underlying mechanisms.

## 1. Introduction

Aging is a natural process in human life that signifies the gradual decline in living organisms. Personal factors such as genetics, race, and diet influence the timing and extent of aging [1]. However, as individuals age and are exposed to environmental stressors such as pollutants, ultraviolet (UV) radiation, smoking, and dietary factors, the levels of reactive oxygen species (ROS) increase. This leads to inflammation and subsequently causes protein and lipid peroxidation, the breakdown of defense systems, and genetic mutations, ultimately accelerating skin pigmentation, aging, and cell apoptosis, potentially leading to the development of skin diseases [2]. Moreover, significant reductions in the quantity and quality of collagen result in age-related changes such as decreased skin elasticity and firmness [3]. Therefore, mitigating issues related to cell aging, cell apoptosis, and extracellular matrix (ECM) synthesis, particularly concerning collagen, elastin, and glycosaminoglycans (GAGs), caused by excessive ROS is effective for the advancement of anti-aging treatments [4].

Peptides are short chains composed of 2 to 50 amino acids linked by peptide bonds [5,6,7]. Amino acids are the building blocks of proteins, and when they are connected in chains, they form peptides [5,6]. Among major bioactive compounds, peptides have attracted scientific attention due to their ideal functional roles as regulators/signaling molecules for homeostasis, stress, immunity, defense, growth, and reproduction [5,6,7]. They also possess strengths such as high safety, low irritation, and cost-effective production [5]. These peptides are derived from various natural sources, including plants, animals, and microorganisms, and have demonstrated diverse physiological effects such as antioxidant, anti-aging, moisturizing, collagen-promoting, and wound-healing properties, as evidenced by various in vitro and in vivo studies, and clinical trial results [5]. For instance, palmitoyl pentapeptide-3 was one of the first synthetic bioactive peptides, stimulating collagen synthesis for anti-aging and wound-healing treatments [8]. Copper Gly-His-Lys (Cu-GHK) was developed for skin healing, collagen synthesis promotion, and DNA damage repair, and is included in cosmetics [9,10]. Acetyl hexapeptide-3 (Argireline) is another popular commercial peptide known for its anti-wrinkle and moisturizing properties [11].

Loricrin is a type of terminally differentiating structural protein that constitutes 70% of the cornified skin outer barrier and is known to contribute to the protective function of the stratum corneum. In vivo, Loricrin is expressed in all mammalian stratified epithelia, with the highest levels found in moist tissues such as the neonatal epidermis, epithelial lining of the oral and anal mucosa, esophagus, foreskin, vagina, and epidermal parts of sweat glands. Notably, Loricrin is utilized as a marker in the early stages of potentially malignant disorders such as oral submucous fibrosis and leukoplakia [12]. 

In a previous study [13], 42 peptides based on the sequence of the skin structural protein Loricrin were identified and synthesized. The elastase inhibitory activity and skin cell proliferating–promoting efficacy of these tetrapeptides were evaluated. As a result, Tetrapeptide-68 (GQVS, MW: 317.18049, Figure 1) was selected as the final anti-wrinkle peptide due to its elastase inhibitory activity and skin cell proliferating–promoting ability. This study aims to evaluate the efficacy of Tetrapeptide-68 (100 ppm) in an O/W formulation for improving periorbital wrinkles in adult women.

## 2. Materials and Methods

### 2.1. Preparation of the O/W Formulation

The product was provided by Sehwa P&C Co., Ltd. (Osong, Chungbuk, Republic of Korea), who manufactured test products containing 100 ppm of Tetrapeptide-68 in an O/W formulation, as well as control products without the aforementioned ingredients. According to the International Nomenclature of Cosmetic Ingredients (INCI), the ingredients of the test product were as follows: disodium EDTA, cetearyl olivate, sorbitan olivate, glycerin, 1,2-hexanediol, carbomer, glyceryl stearate, cetearyl glucoside, glucose, cetearyl alcohol, caprylic/capric triglyceride, cetyl ethylhexanoate, hydrogenated polydecene, shea butter, tromethamine, ethylhexylglycerin, butylene glycol, vinyl dimethicone, polyglyceryl-10 stearate, and hydrogenated lecithin. The control products contained all these same ingredients, except for Tetrapeptide-68.

### 2.2. Trial Design

The study was designed to evaluate the efficacy of a specific cosmetic product on 25 individuals aged 30 to 65 years with naturally aged skin over a period of 12 weeks. The test product was applied from day one to week twelve. Participants were instructed to apply an appropriate amount of the product evenly to their skin every morning and evening after cleansing. The researchers measured the participants’ skin physiological parameters at baseline (the day before day 1), week 4, week 8, and week 12. This study was conducted in accordance with Korean Good Clinical Practice (KGCP) guidelines and relevant regulations and guidelines of the Ministry of Food and Drug Safety (MFDS) in Korea, following approval by the COREDERM Co., Ltd. Institutional Review Board (Seoul, Republic of Korea, IRB approval No. CDIRB-RR-21-003).

### 2.3. Study Participants

All participants were recruited from the Corederm Skin Science Research Center, Co., Ltd. (Seoul, Republic of Korea). A total of 30 Asian women aged between 30 and 65 years, who had just begun to develop or already had developed wrinkles around the eyes, were enrolled in the study according to inclusion and exclusion criteria. A total of 5 subjects dropped out of the study for personal reasons, and 25 subjects completed the study. The inclusion criteria were as follows: women aged 30 to 65 years who had just begun to develop or already had wrinkles around the eyes (visible assessment grade 3 or higher, according to the Corederm Standard Operating Procedure (SOP)); those who had been sufficiently informed by the researchers about the purpose and content of the study and voluntarily signed the informed consent form; and those who were available for follow-up during the study period. The exclusion criteria were as follows: those who were pregnant, breastfeeding, or planning to become pregnant within 6 months; those with psychiatric disorders or infectious skin diseases; those with chronic wasting diseases (e.g., asthma, diabetes, hypertension, hyperthyroidism, or hypothyroidism); those using topical skin treatments containing antibiotics, immunosuppressants, or steroids for more than one month on the test area; those taking contraceptives, antihistamines, or anti-inflammatory drugs; those with skin diseases, skin allergies, sensitive or hypersensitive skin, or atopic dermatitis; those with abnormal skin findings such as moles, acne, telangiectasia, erythema, or scars on the test area; those consuming drugs or foods claiming to have anti-wrinkle effects; those who used similar or identical cosmetics or pharmaceuticals on the test area within 3 months prior to the study; those who underwent dermatological procedures (e.g., skin peeling, wrinkle removal) on the test area within 6 months prior to the study; those who had severe irritation or allergies to cosmetics, pharmaceuticals, or sunlight exposure; those who participated in a similar study within the last 6 months; employees of the clinical research institute; and those who were deemed unsuitable for the study by the principal investigator.

### 2.4. Visual Assessment of Wrinkles around the Eyes

In this study, an imaging device, VISIA-CR 2.3 (Canfield Scientific Inc., Parsippany-Troy Hills, NJ, USA), was used to photograph the periocular area under consistent conditions regarding posture, distance, and illumination before Tetrapeptide-68 cream use, and at 4, 8, and 12 weeks of Tetrapeptide-68 cream use. The periocular area was photographed in both optical and polarized modes. After capturing the images, two dermatologists independently evaluated the wrinkles around the eyes on a 10-point scale (0 = no wrinkles, 10 = many wrinkles), and the average of their assessments was analyzed.

### 2.5. Instrumental Assessment of Wrinkles around the Eyes

#### 2.5.1. Preparation of Wrinkle Replicas

To measure wrinkles around the eyes, replicas reflecting the actual state of skin wrinkles were made using the Skin Visiometer SV Replica full kit (Courage-Khazaka Electronic, Köln, Germany). During the creation of the replicas, the periocular area of the subjects was shaved and cleaned. The subjects laid at a 135° angle while a mixture of Replica Solution B and Solution C was applied in a fixed amount to a production film and attached to the crow’s feet area, capturing the skin’s shape exactly as it was. Replicas of the left and right periocular wrinkle areas were created before Tetrapeptide-68 cream use, and at 4, 8, and 12 weeks of Tetrapeptide-68 cream use.

#### 2.5.2. Skin Visiometer SV700 Measurements

At each evaluation point, the replicas were measured using the Skin Visiometer SV700 (Courage-Khazaka Electronic, Cologne, Germany) to assess the parameters of R1 (skin roughness), R2 (maximum roughness), R3 (average roughness), R4 (smoothness depth), and R5 (arithmetic average roughness). The Skin Visiometer SV700 quantifies differences in light transmission based on the thickness of the replicas, which varies according to the skin’s texture. Reflecting this principle, the parameters R1, R2, R3, R4, and R5 were measured in three ways: horizontally, vertically, and circularly. The analysis was conducted using the circle method, which considers the characteristics of wrinkles forming in multiple directions.

#### 2.5.3. PRIMOS Lite Measurements

PRIMOS lite (Canfield Scientific Inc.) is a 3D skin measurement device that uses Digital Micromirror Device (DMD) technology for non-contact digital stripe projection. The captured 3D images enable consistent measurement of the same area through matching and overlay functions. This device can analyze changes in skin wrinkles, pores, surface roughness, area, and volume using various parameters. In this study, the wrinkle conditions of the left and right periocular areas were photographed before product use, and at 4, 8, and 12 weeks of product use. Wrinkles were analyzed using the parameters Rz (average wrinkle depth, μm) and Ra (average roughness, μm). A decrease in these parameter values indicates an improvement in wrinkles.

### 2.6. Questionnaire Evaluation by Study Participants

In this study, participants completed a questionnaire evaluating the Tetrapeptide-68 cream’s efficacy after 4, 8, and 12 weeks of use, and its usability after 12 weeks of use. The efficacy was rated on a 3-point scale (1, no change; 2, improved; 3, much improved), and usability was rated on a 5-point scale (1, very bad; 2, bad; 3, average; 4, good; 5, very good). Positive responses (efficacy scores of 2 or 3/usability scores of 4 or 5) were analyzed.

### 2.7. Skin Safety Evaluation

Skin safety evaluation was conducted by modifying the method of Im et al. [14]. Skin irritation was assessed through interviews and medical examinations at 4, 8, and 12 weeks of product use.

### 2.8. Statistical Analysis

All data were analyzed for statistical significance using the SPSS Package Program ver. 27 (IBM, IL, USA). The normality of the data was tested using the Shapiro–Wilk test and kurtosis/skewness. Each measurement was repeated three times to ensure accuracy and reliability. The data from the product usability questionnaire were analyzed using the chi-square test. A value of *p* < 0.05 was considered statistically significant.

## 3. Results

### 3.1. Characteristics of Participants’ Skin

Twenty-five women aged 40 to 60 years (mean 51.8 ± 4.9) who met the inclusion and exclusion criteria were selected for this study and completed it. The skin characteristics of the participants were surveyed and recorded using a questionnaire. The analysis results are presented in Table 1 and Table 2.

### 3.2. Visual Analysis of Wrinkles around the Eyes

To verify the improvement effect of Tetrapeptide-68 on the formation of wrinkles around the eyes, participants’ wrinkles were independently assessed under specific lighting conditions. Skin damage, as judged by visual evaluation, significantly (*p* < 0.05) recovered after 12 weeks of continuous use of the cream containing Tetrapeptide-68 compared to after using the placebo cream (Figure 2).

### 3.3. Analysis of Wrinkles around the Eyes Using Replica Images

#### 3.3.1. Analysis of Wrinkle Parameters around the Eyes Using Skin Visiometer

For the R1 parameter, the results showed a significant reduction (*p* < 0.05) in the test group after 12 weeks of use compared to before product use, while no significant change was observed in the control group. When comparing the groups, the change in the R1 parameter after 12 weeks was −0.014 ± 0.022 in the test group and 0.006 ± 0.029 in the control group, showing a significant difference (*p* < 0.05). Repeated-measures ANOVA also indicated a significant difference between the two groups (*p* < 0.05, Figure 3a). For the R2 parameter, the results showed a significant reduction (*p* < 0.05) in the test group after 12 weeks of use compared to before product use, while no significant change was observed in the control group. When comparing the groups, the change in the R2 parameter after 12 weeks was −0.005 ± 0.010 in the test group and 0.003 ± 0.015 in the control group, showing a significant difference (*p* < 0.05). Repeated-measures ANOVA also indicated a significant difference between the two groups (*p* < 0.05, Figure 3b). For the R3 parameter, the results showed a significant reduction (*p* < 0.05) in the test group after 12 weeks of use compared to before product use, while no significant change was observed in the control group. When comparing the groups, the change in the R3 parameter did not show a significant difference at any time point, but repeated-measures ANOVA indicated a significant difference between the two groups (*p* < 0.05, Figure 3c). For the R4 parameter, the results showed a significant reduction (*p* < 0.05) in the test group after 4, 8, and 12 weeks of use compared to before product use, while no significant change was observed in the control group. When comparing the groups, the change in the R4 parameter after 12 weeks was −0.010 ± 0.015 in the test group and 0.006 ± 0.016 in the control group, showing a significant difference (*p* < 0.05). Repeated-measures ANOVA also indicated a significant difference between the two groups (*p* < 0.05, Figure 3d). For the R5 parameter, the results showed a significant reduction (*p* < 0.05) in the test group after 12 weeks of use compared to before product use, while no significant change was observed in the control group. When comparing the groups, the change in the R5 parameter after 12 weeks was −0.004 ± 0.007 in the test group and 0.002 ± 0.009 in the control group, showing a significant difference (*p* < 0.05). Repeated-measures ANOVA also indicated a significant difference between the two groups (*p* < 0.05, Figure 3e).

#### 3.3.2. Analysis of Crow’s Feet Wrinkle Parameters Using Primos Lite

For the Rz parameter, the results showed a significant reduction (*p* < 0.05) in the test group after 4, 8, and 12 weeks of use compared to before product use, while no significant change was observed in the control group. When comparing the groups, the change in the Rz parameter after 12 weeks was −7.40 ± 9.53 in the test group and −0.98 ± 5.71 in the control group, showing a significant difference (*p* < 0.05). Repeated-measures ANOVA also indicated a significant difference between the two groups (*p* < 0.05, Figure 4a). For the Ra parameter, the results showed a significant reduction (*p* < 0.05) in the test group after 4, 8, and 12 weeks of use compared to before product use, while no significant change was observed in the control group. When comparing the groups, the change in the Ra parameter after 12 weeks was −1.68 ± 2.06 in the test group and −0.24 ± 1.40 in the control group, showing a significant difference (*p* < 0.05). Repeated-measures ANOVA also indicated a significant difference between the two groups (*p* < 0.05, Figure 4b). In the 3D images, it was confirmed that both deep and fine wrinkles around the eyes were improved in the test group (Figure 4c).

### 3.4. Questionnaire Evaluation

#### 3.4.1. Evaluation of the Efficacy Questionnaire

For the ‘improvement in wrinkles around the eyes’ item, the test group reported 68% after 4 weeks, 92% after 8 weeks, and 84% after 12 weeks. The control group reported 72% after 4 weeks, 88% after 8 weeks, and 80% after 12 weeks. For the ‘reduction in fine wrinkles’ item, the test group reported 72% after 4 weeks, 84% after 8 weeks, and 92% after 12 weeks. The control group reported 68% after 4 weeks, 80% after 8 weeks, and 88% after 12 weeks. For the ‘skin improvement’ item, the test group reported 92% after 4 weeks, 96% after 8 weeks, and 92% after 12 weeks. The control group reported 96% after 4, 8, and 12 weeks. Positive response rates were observed, but there were no significant differences between the two groups for any of the questionnaire items (Figure 5).

#### 3.4.2. Product Usability Questionnaire

Results from the usability questionnaire regarding the product’s usability showed that after 12 weeks of use, both the test and control groups reported 48% satisfaction with ‘scent satisfaction’. For ‘color satisfaction’, both groups reported 76%. Regarding ‘absorption satisfaction’, the test group reported 84% satisfaction while the control group reported 68%. In terms of ‘overall satisfaction’, the test group reported 84% satisfaction while the control group reported 80%. Positive response rates were observed for all questionnaire items, but there were no significant differences between the two groups (Figure 6).

### 3.5. Skin Safety Evaluation

No skin adverse reactions were observed in any of the study participants during the study period (Table 3).

## 4. Discussion

Due to the increasing demand for cosmeceutical products, there is growing interest in the development of next-generation products based on bioactive peptides. Therefore, the safety evaluation of cosmetic peptides, especially from an efficacy perspective, must be seriously considered. It is well known that an effective amount of cosmetic peptides can provide significant positive effects on the skin, while minimizing undesirable effects such as skin toxicity, sensitivity, and irritation [15]. Bioactive peptides are widely used as functional ingredients in dermatology and the cosmetic fields due to their potent single/multi-functional biological properties, such as antimicrobial, antioxidant, anti-aging, and anti-inflammatory activities. These peptides can enhance skin health by improving various aspects including extracellular matrix synthesis, innate immunity, inflammation, and pigmentation disorders [16,17,18,19].

Skin aging is commonly manifested by increased wrinkle formation, laxity, and sagging of the skin. These changes have been suggested to be promoted by various factors such as chronic muscle contraction or gravity, primarily involving quantitative reduction and qualitative deterioration of the extracellular matrix (ECM) supporting the dermis [20]. Decreased expression of ECM proteins and increased activity of ECM-degrading enzymes, including various matrix metalloproteinases (MMPs), leading to structural damage of filaments, have been proposed as fundamental mechanisms underlying ECM damage. This has been a major target of traditional anti-aging strategies. Numerous studies have been conducted on cosmetic ingredients that stimulate skin collagen synthesis, including ascorbic acid [21], retinol [22], natural extracts [23], growth factors [24], and collagen or its degraded forms [25]. The use of MMP inhibitors or the downregulation of MMP expression to inhibit ECM degradation have also been proposed as anti-aging therapies [26]. In recent years, the anti-aging potential of naturally derived materials has also been investigated, showing advancements in skincare applications that enhance skin health and potentially offer new avenues for cosmetic development [27,28,29,30].

Tetrapeptide-68 has been identified as a promising anti-aging agent. Previous research has demonstrated that Tetrapeptide-68 exhibits elastase inhibitory activity, which helps in reducing the breakdown of elastin, a key protein responsible for skin elasticity and firmness. Additionally, Tetrapeptide-68 has been shown to enhance skin cell proliferation, promoting the regeneration of skin cells and contributing to the reduction in fine lines and wrinkles [13]. The peptide’s anti-aging effects are attributed to its ability to support the synthesis and maintenance of key components of the ECM, such as collagen and elastin, thereby improving skin elasticity and firmness.

In this study, it was found through visual assessment that skin wrinkles significantly decreased after applying Tetrapeptide-68 cream for 4, 8, and 12 weeks compared to a placebo cream. Interestingly, Tetrapeptide-68 cream notably reduced wrinkles compared to the placebo cream after 12 weeks of visual evaluation (*p* < 0.05, Figure 2). The application of skincare products can potentially affect overall exposure and increase the risk of local side effects such as sensitization, photoreactivity, and irritation; hence, safety evaluations are necessary to assess the inherent risks of ingredients included in the product [31]. While adverse reactions to personal care products or cosmetics may vary individually, cosmetic ingredients are generally considered safe [31]. In accordance with the functional cosmetics efficacy evaluation guidelines from the South Korean Ministry of Food and Drug Safety (MFDS), skin safety evaluations were conducted on more than 20 women in this study. Further studies with larger sample sizes, extended periods of product use, longer follow-up durations, and testing with various concentrations are needed. However, in this study, no dermatological adverse reactions were observed in any of the subjects (Table 3).

According to the analysis, compared to before Tetrapeptide-68 cream use, the R4 parameter showed significant improvement after 4, 8, and 12 weeks of use, while R1, R2, R3, and R5 parameters demonstrated significant improvement after 12 weeks (*p* < 0.05). When comparing between groups, significant differences were observed in the changes in R1, R2, R4, and R5 parameters after 12 weeks compared to the control group (*p* < 0.05), as indicated by repeated-measures ANOVA results (*p* < 0.05).

According to the analysis, compared to before Tetrapeptide-68 cream use, the Rz and Ra parameters showed significant improvement after 4, 8, and 12 weeks of use (*p* < 0.05). When comparing between groups, significant differences were observed in the changes in Rz and Ra parameters after 12 weeks compared to the control group (*p* < 0.05), as indicated by repeated-measures ANOVA results (*p* < 0.05).

However, the visual analysis by dermatologists showed a significant change in the test group as early as 4 weeks. This significant change was particularly evident in the R4, Rz, and Ra parameters. These early improvements could be attributed to the initial rapid response of the skin to Tetrapeptide-68, reflecting changes in skin texture and smoothness that are visually noticeable before other parameters show significant changes. Despite the visual analysis by the dermatologists and mechanical measurements using skin replicas indicating significant improvements, the test group subjects reported similar scores for “improvement in wrinkles around the eyes” compared to the control group. This discrepancy suggests that the improvements may be more pronounced at the microscopic level and thus not as easily recognized by the participants themselves.

Evaluation of the efficacy questionnaire, regarding the ‘improvement in wrinkles around the eyes’ item, the experimental group showed positive response rates of 68% after 4 weeks, 92% after 8 weeks, and 84% after 12 weeks, while the control group showed 72% after 4 weeks, 88% after 8 weeks, and 80% after 12 weeks. For the ‘reduction in fine lines’ item, the experimental group showed positive response rates of 72% after 4 weeks, 84% after 8 weeks, and 92% after 12 weeks, whereas the control group showed 68% after 4 weeks, 80% after 8 weeks, and 88% after 12 weeks. Regarding the ‘overall skin improvement’ item, the experimental group demonstrated positive response rates of 92% after 4 weeks, 96% after 8 weeks, and 92% after 12 weeks, while the control group showed 96% after 4 weeks, 8 weeks, and 12 weeks. However, there were no significant differences observed between the two groups in all survey items.

Evaluation of the efficacy questionnaire, regarding the ‘improvement in wrinkles around the eyes’ item, the Tetrapeptide-68 cream group showed positive response rates of 68% after 4 weeks, 92% after 8 weeks, and 84% after 12 weeks, while the control group showed 72% after 4 weeks, 88% after 8 weeks, and 80% after 12 weeks. For the ‘reduction in fine wrinkles, the Tetrapeptide-68 cream group showed positive response rates of 72% after 4 weeks, 84% after 8 weeks, and 92% after 12 weeks, whereas the control group showed 68% after 4 weeks, 80% after 8 weeks, and 88% after 12 weeks. Based on these results, Tetrapeptide-68 cream is considered to be a product with wrinkle-improving effects on human skin.

## 5. Conclusions

In this study, we evaluated the wrinkle-improving effects and skin safety of Tetrapeptide-68 cream. Ultimately, Tetrapeptide-68 cream effectively improves wrinkles associated with skin aging and shows promising potential as a cosmeceutical product that can be safely used without causing skin irritation. This is expected to contribute significantly to the development of cosmetics based on bioactive peptides.

## Figures and Tables

**Figure 1 pharmaceutics-16-00987-f001:**
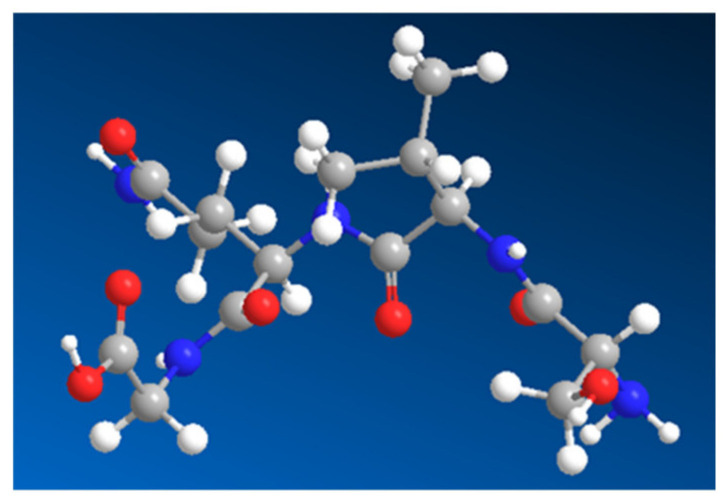
Chemical 3D structure of Tetrapeptide-68 (GQVS). The image represents the three-dimensional structure of Tetrapeptide-68, which is composed of the amino acids Glycine (G), Glutamine (Q), Valine (V), and Serine (S). The color-coded atoms are as follows: Gray spheres represent carbon (C) atoms. White spheres represent hydrogen (H) atoms. Red spheres represent oxygen (O) atoms. Blue spheres represent nitrogen (N) atoms.

**Figure 2 pharmaceutics-16-00987-f002:**
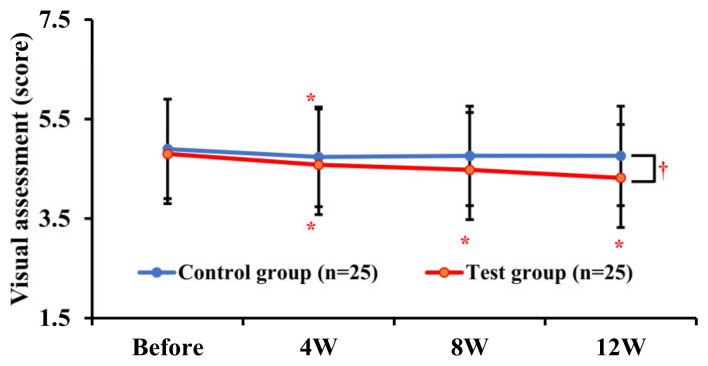
Changes in visual parameters after 12 consecutive weeks of treatment with test and control groups. Data are shown as the mean ± SEM (*n* = 25, each group), * *p* < 0.05 vs. before treatment, ^†^ *p* < 0.05 vs. control group. Test group: Tetrapeptide-68 cream; control group: placebo cream.

**Figure 3 pharmaceutics-16-00987-f003:**
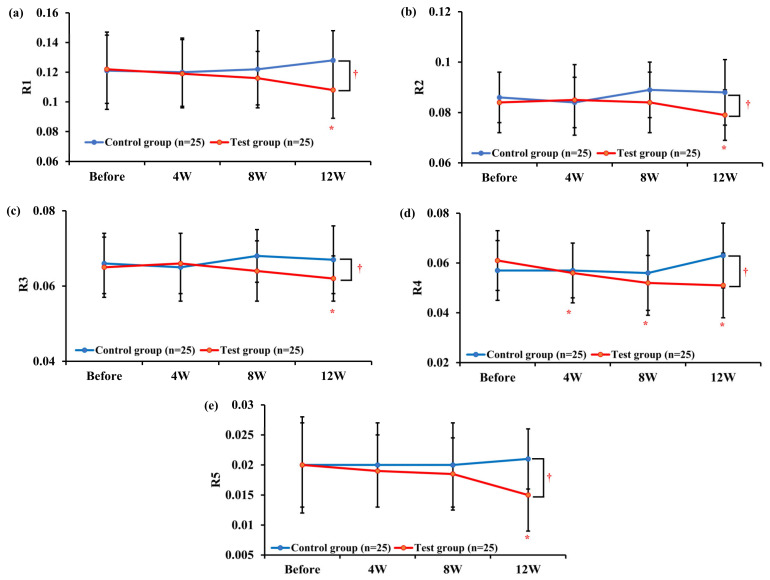
Changes in skin parameters after treatment for 12 weeks in the test and control groups. (**a**) R1; Changes in Skin Roughness. (**b**) R2; Changes in Maximum Roughness. (**c**) R3; Changes in Average Roughness. (**d**) R4; Changes in Smoothness Depth. (**e**) R5; Changes in Arithmetic Average Roughness. Data are shown as the mean ± SEM (*n* = 25, each group), * *p* < 0.05 vs. before treatment, ^†^ *p* < 0.05 vs. control group. Test group: Tetrapeptide-68 cream; control group: placebo cream.

**Figure 4 pharmaceutics-16-00987-f004:**
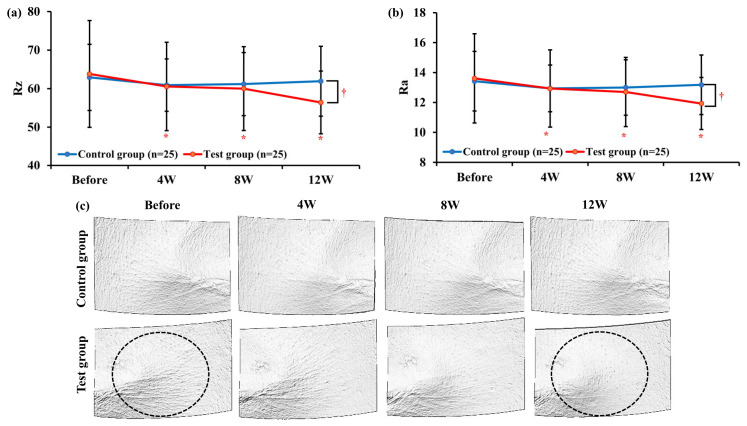
Changes in skin parameters after treatment for 12 weeks in the test and control groups. (**a**) Rz; Changes in average roughness. (**b**) Ra; Changes in arithmetic average roughness. (**c**) 3D image of wrinkles around the eyes. The improved areas are highlighted with circles. Data are shown as the mean ± SEM (*n* = 25, each group), * *p* < 0.05 vs. before treatment, ^†^ *p* < 0.05 vs. control group. Test group: Tetrapeptide-68 cream; control group: placebo cream.

**Figure 5 pharmaceutics-16-00987-f005:**
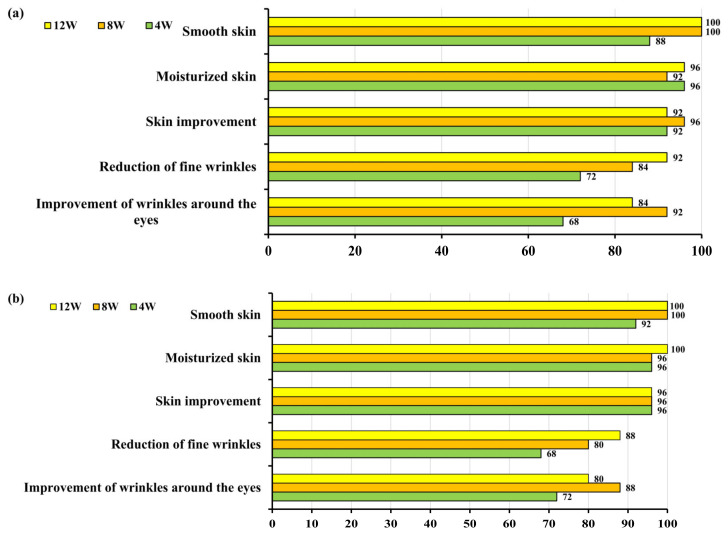
Survey results on the effectiveness of treatment over 12 weeks in the test and control groups. (**a**) Test group: Tetrapeptide-68 cream; (**b**) control group: placebo cream.

**Figure 6 pharmaceutics-16-00987-f006:**
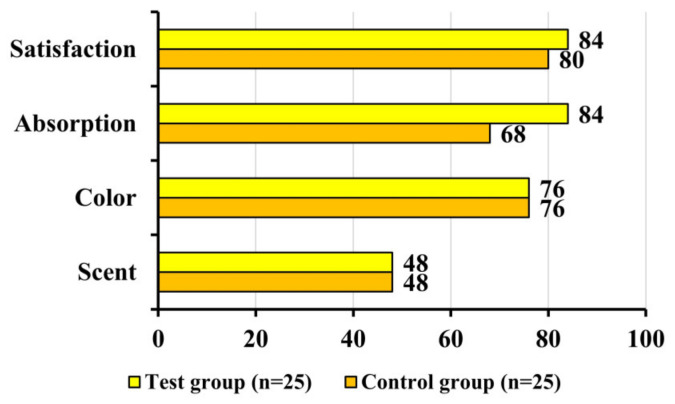
Comparative sensorial profile of the test and control groups in terms of usability (positive answers, %). Test group: Tetrapeptide-68 cream; control group: placebo cream.

**Table 1 pharmaceutics-16-00987-t001:** Characteristics of participants’ skin (*n* = 25).

Item	Classification	Frequency (N)	Percentage (%)
Age	40s	6	24.00
	50s	18	72.00
	60s	1	4.00
Skin type	Dry	16	64.00
	Normal	6	24.00
	Oily	0	0.00
	Dry and oily	3	12.00
	Problematic	0	0.00

**Table 2 pharmaceutics-16-00987-t002:** Skin condition of participants according to physiological factors (*n* = 25).

Item	Classification	Frequency (N)	Percentage (%)
Hydration	Sufficient	0	0.00
	Normal	10	40.00
	Deficient	14	56.00
	Very deficient	1	4.00
Sebum	Very glossy	1	4.00
	Normal	18	72.00
	Deficient	6	24.00
Dryness	Moist	0	0.00
	Normal	8	32.00
	Dry	15	60.00
	Very dry	2	8.00
Number of showers (per week)	Less than 1 time	0	0.00
	2–3 times	5	20.00
	4–6 times	12	48.00
	Once daily	7	28.00
	More than twice a day	1	4.00
Use body products	Not used	3	12.00
	Used occasionally	18	72.00
	Always used	4	16.00
UV exposure (per day)	Less than 1h	6	24.00
	1–3 h	17	68.00
	More than 3 h	2	8.00
Sleeping h (per day)	Less than 5 h	3	12.00
	5–8 h	22	88.00
	More than 8 h	0	0.00
Smoking (per day)	No	25	100.00
	Less than 10 pieces	0	0.00
	More than 10 pieces	0	0.00
Irritability	Yes	1	4.00
	No	24	96.00
Stinging	Yes	0	0.00
	No	25	100.00
Adverse reaction	Yes	0	0.00
	No	25	100.00
Skin changes during menstruation	Yes	2	8.00
	No	10	40.00
	Not applicable	13	52.00
Menstrual cycle	One week before menstruation	2	8.00
	During menstruation	0	0.00
	Within one week after menstruation	5	20.00
	Other	5	20.00
	Not applicable	13	52.00

**Table 3 pharmaceutics-16-00987-t003:** Evaluation of skin safety (adverse reaction) results (*n* = 25).

Symptom		4 W	8 W	12 W
Subjective irritation	Itching	0	0	0
	Stinging	0	0	0
	Tickling	0	0	0
	Burning	0	0	0
	Prickling	0	0	0
	Stiffness	0	0	0
	Tightening	0	0	0
	Other	0	0	0
Objective irritation	Erythema	0	0	0
	Edema	0	0	0
	Scale	0	0	0
	Papule	0	0	0
	Other	0	0	0

## Data Availability

The data presented in this study are available in this article.

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
