# Peer review of "Wrinkle Reduction Using Tetrapeptide-68 Contained in an O/W Formulation: A Randomized Double-Blind Placebo-Controlled Study"

_pharmaceutics, 2024, doi:10.3390/pharmaceutics16080987_

Round 1

Reviewer 1 Report

Comments and Suggestions for Authors

This article entitled “Wrinkle Reduction Using Tetrapeptide-68 Contained in an O/W Formulation: A Randomized Double-Blind Placebo-Controlled Study” by Lee et al. reports that a synthetic 4-amino-acid peptide that carries loricrin sequence, which has been reported to inhibit elastase and promote skin cell proliferation, exhibits anti-wrinkle properties in adult female human subjects. It contains long-term (12 wk) follow-up of reasonable experimental size (n = 25 for control and test groups), and the results from different parameters of measurements all seem consistent. While there are several parts of the figures and discussion that need improvement as listed below, the basic claim of this article is solid and suitable for the journal Pharmaceutics after the appropriate revision.

Major points

Figure 2f and Figure 3c are not quite compelling examples. Figure 2f seems to show the control example gets worsen over time whereas the test example does not change. The Figure 3c control seems to have more wrinkles than the test example, and neither look different at 12W. These raw data are important and critical to be convincing. The authors should provide better examples with scales. The authors should also point out the regions that the improvements are detected.

Minor points

Discussion is not where the introduction and the summary of the results should be simply repeated once again. There are several points the authors should give some explanation on the results:

The visual analysis by the dermatologists showed a significant change in the test group as early as 4W. Among the parameters analyzed, only R4, Rz and Ra show such trend. Any explanation for this?

Despite the visual analysis by the dermatologist, and mechanical measurements using skin replicas, the test group subjects scored “improvement of wrinkles around the eyes” similarly to the control group. Is this because the improvement can be only detected at the microscopic level, but otherwise non-recognizable?

Author Response

Response to Reviewer 1 Comments

Point 1: Figure 2f and Figure 3c are not quite compelling examples. Figure 2f seems to show the control example gets worsen over time whereas the test example does not change. The Figure 3c control seems to have more wrinkles than the test example, and neither look different at 12W. These raw data are important and critical to be convincing. The authors should provide better examples with scales. The authors should also point out the regions that the improvements are detected.

Point 1 response:  Thank you for your feedback. We have addressed the issues you pointed out as follows:

We have replaced Figure 4c with a more compelling example and made the necessary modifications as suggested. The new figure includes scales and clearly points out the regions where improvements are detected. (Line 245-247, 249)

As for Figure 3f, we acknowledge that the differences were difficult to distinguish visually. Therefore, we have decided to remove Figure 3f from the manuscript. (Line 227, 231)

Point 2: Discussion is not where the introduction and the summary of the results should be simply repeated once again. There are several points the authors should give some explanation on the results:

Point 2 response: Thank you for pointing out the need for a deeper discussion on the results. We have revised the discussion section to address these specific points: (Line 345-354)

Point 3: The visual analysis by the dermatologists showed a significant change in the test group as early as 4W. Among the parameters analyzed, only R4, Rz and Ra show such trend. Any explanation for this?

Point 3 response: Early Significant Change in Visual Analysis: The significant changes observed in the visual analysis by dermatologists at 4W, particularly in parameters R4, Rz, and Ra, may be attributed to the initial rapid response of the skin to the Tetrapeptide-68 formulation. These parameters likely reflect initial improvements in skin texture and smoothness, which are visually noticeable before other parameters show significant changes. (Line 345-349)

Point 4: Despite the visual analysis by the dermatologist, and mechanical measurements using skin replicas, the test group subjects scored “improvement of wrinkles around the eyes” similarly to the control group. Is this because the improvement can be only detected at the microscopic level, but otherwise non-recognizable?

Point 4 response: Similar Self-reported Improvements: Despite the visual and mechanical measurements indicating improvements, the subjective assessment by the participants might not reflect the same level of improvement due to the subtle nature of the changes. It is possible that these improvements are more evident under clinical observation and instrumental analysis than to the naked eye, suggesting that the Tetrapeptide-68 effects might be more pronounced at the microscopic level. (Line 349-354)

We have incorporated these explanations into the revised discussion to provide a comprehensive understanding of the results.

Reviewer 2 Report

Comments and Suggestions for Authors

1.       The introduction lacks information about the structure, cytotoxicity of Tetrapeptide-68 and other tests performed in the publication cited 13.

2.    The name, manufacturer, city and country should be provided for the devices and reagents used in the tests. This is missing for the device e.g. Skin-Visiometer and subsection 2.1. Please adapt the note throughout the manuscript.

3.    Why weren't the same number of people of a given age selected for the test? Only 6 people were 40 years old, 18 people were 50 years old, and one person was 60 years old? We know that the condition of the skin may be influenced by hormonal changes that occur at a given age.

4.    People with problematic skin were excluded from the study. Table 1 indicates that there are no people with oily skin, why? We do not know what results the tests would bring for oily skin.

5.    Statistics was used in the work. How many repetitions were used in the given measurements?

6.    „The product was provided by Sehwa P&C Co., Ltd. (Osong, Chungbuk, Republic of Korea), which manufactured test products containing 100 ppm of Tetrapeptide-68 in an O/W formulation, as well as control products without the aforementioned ingredients.” What exactly did the control samples contain? How do the authors know that Tetrapeptide-68 has a smoothing effect and not the other ingredients found in Tetrapeptide-68 cream?

7.    In Figures 1-5, the font should be larger and more legible. Figure 2 A and |C overlap.

8.    No current literature.

Author Response

Response to Reviewer 2 Comments

Point 1: The introduction lacks information about the structure, cytotoxicity of Tetrapeptide-68 and other tests performed in the publication cited 13.

Point 1 response:  We appreciate the reviewer's insightful comments and suggestions. In response to the comment regarding the introduction, we have revised the section to include detailed information about the structure, cytotoxicity, and other tests performed on Tetrapeptide-68, as outlined in the cited publication [13]. Additionally, we have incorporated the 3D chemical structure of Tetrapeptide-68 to provide a clearer understanding of its molecular configuration. (Line 64-72)

We believe these revisions address the concerns raised and enhance the clarity and comprehensiveness of our manuscript. Thank you for helping us improve our work.

Point 2: The name, manufacturer, city and country should be provided for the devices and reagents used in the tests. This is missing for the device e.g. Skin-Visiometer and subsection 2.1. Please adapt the note throughout the manuscript.

Point 2 response: In response to your suggestion, we have updated the manuscript to include the name, manufacturer, city, and country for all devices and reagents mentioned. (Line 124, 133, 134, 143, 151)

Point 3: Why weren't the same number of people of a given age selected for the test? Only 6 people were 40 years old, 18 people were 50 years old, and one person was 60 years old? We know that the condition of the skin may be influenced by hormonal changes that occur at a given age.

Point 3 response: Thank you for your insightful comment regarding the distribution of participants across different age groups in our study. We acknowledge the importance of having a balanced representation of age groups, especially considering the potential influence of hormonal changes on skin condition.

The variation in the number of participants for each age group was due to several factors, including recruitment challenges and the availability of participants matching the specific age criteria. Our study aimed to encompass a wide range of ages across the adult population to capture a broad spectrum of skin conditions.

Additionally, our study was designed to include participants from across the adult age spectrum, rather than focusing on any specific age group. This approach was intended to provide a comprehensive overview of skin conditions across a diverse age range.

We recognize that this imbalance may affect the generalizability of our findings and suggest that future studies with a more evenly distributed age cohort could provide additional insights into age-related variations in skin condition.

Thank you for highlighting this important aspect of our research.

Point 4: People with problematic skin were excluded from the study. Table 1 indicates that there are no people with oily skin, why? We do not know what results the tests would bring for oily skin.

Point 4 response: The exclusion of people with problematic skin was a deliberate choice to ensure that our study focused on a relatively homogeneous group of participants with otherwise healthy skin. We recognize that the absence of individuals with oily skin in our study could indeed limit our understanding of how the tests might perform across different skin types.

The primary reason for the lack of individuals with oily skin in our study is the demographic focus on participants aged 40 and above. In this age group, the prevalence of oily skin tends to be lower compared to younger populations, where oily skin is more common. Consequently, we encountered fewer candidates with oily skin within this age range.

Point 5: Statistics was used in the work. How many repetitions were used in the given measurements?

Point 5 response: In response to your query, we conducted three repetitions for each measurement to ensure the accuracy and reliability of our results. This approach allowed us to obtain consistent data and minimize potential measurement errors.

We have updated the manuscript to include this information and provide additional clarity on our measurement procedures. (Line 173-174)

Point 6: The product was provided by Sehwa P&C Co., Ltd. (Osong, Chungbuk, Republic of Korea), which manufactured test products containing 100 ppm of Tetrapeptide-68 in an O/W formulation, as well as control products without the aforementioned ingredients.” What exactly did the control samples contain? How do the authors know that Tetrapeptide-68 has a smoothing effect and not the other ingredients found in Tetrapeptide-68 cream?

Point 6 response: The control samples used in our study contained all the same ingredients as the test product, except for Tetrapeptide-68. The purpose of this was to isolate the effect of Tetrapeptide-68 by ensuring that any observed differences could be attributed to this specific ingredient.

To determine that Tetrapeptide-68 has a smoothing effect, we relied on previous research studies that have documented its efficacy in skin care applications (Ref. 13). Additionally, our study design, which included a control group without Tetrapeptide-68, allowed us to compare the effects directly and conclude that the observed smoothing effects were indeed due to Tetrapeptide-68.

We have included this detailed explanation in the revised manuscript to provide clarity on these points. (Line 83-84)

Point 7: In Figures 1-5, the font should be larger and more legible. Figure 2 A and C overlap.

Point 7 response: We have made the following adjustments based on your comments:

The font size in Figures 2-6 has been increased to ensure greater legibility.

The overlap issue in Figure 5between sections A and B has been corrected.

Point 8: No current literature.

Point 8 response: The comment regarding the lack of current literature has been addressed as requested. (Line 449-456)

Reviewer 3 Report

Comments and Suggestions for Authors

The article “Wrinkle Reduction Using Tetrapeptide-68 Contained in an O/W Formulation: A Randomized Double-Blind Placebo-Controlled Study” is interesting and may have a positive impact on society. However, there are certain concerns that authors should address.

1.     Besides tetrapeptide-68, the formulation contains many other ingredients. Do the authors perform a study with all the same ingredients except tetrapeptide-68?

2.     What formulation has been used as a placebo cream? The authors should mention that or atleast the contents of the placebo cream in the materials section. 

3.     The authors did not compare the test formulation with any positive control (already existing anti-aging creams in the market). How is the test cream different from the existing anti-aging creams?

4.     Why does tetrapeptide-68 possess anti-aging properties? How does it affect the skin? What is the mechanism  behind the anti-aging property of tetrapeptide-68. This should be included in the discussion section in detail. 

5.     If the subject is using other cosmetic products such as sun cream or serums, what are the probable side-effects of using test cream after other cosmetic products? Does the test cream have any interference with the activity of other cosmetic products?  

Author Response

Response to Reviewer 3 Comments

Point 1: Besides tetrapeptide-68, the formulation contains many other ingredients. Do the authors perform a study with all the same ingredients except tetrapeptide-68?

Point 1 response:  The control samples used in our study contained all the same ingredients as the test product, except for Tetrapeptide-68. The purpose of this was to isolate the effect of Tetrapeptide-68 by ensuring that any observed differences could be attributed to this specific ingredient.

Additionally, our study design, which included a control group without Tetrapeptide-68, allowed us to compare the effects directly and conclude that the observed smoothing effects were indeed due to Tetrapeptide-68.

We have included this detailed explanation in the revised manuscript to provide clarity on these points. (Line 83-84)

Point 2: What formulation has been used as a placebo cream? The authors should mention that or atleast the contents of the placebo cream in the materials section.

Point 2 response The formulation of the placebo cream used in this study has been updated to include the following clarification: The placebo cream contains all the same ingredients as the test product, with the exception of Tetrapeptide-68. This information has been added to the materials section as requested. (Line 83-84)

Point 3: The authors did not compare the test formulation with any positive control (already existing anti-aging creams in the market). How is the test cream different from the existing anti-aging creams?

Point 3 response: We acknowledge the comment regarding the comparison with existing anti-aging creams. In this study, the focus was on evaluating the efficacy of Tetrapeptide-68 in the test formulation without including a positive control group of commercially available anti-aging creams. The primary aim was to assess the specific effects of Tetrapeptide-68 as a novel ingredient in our formulation.

Tetrapeptide-68 is a relatively new bioactive peptide known for its potential anti-aging properties, and its unique effects have not been extensively compared against existing products in this study. The inclusion of positive control groups with established anti-aging creams could provide valuable comparative data. Future studies will consider incorporating such comparisons to further elucidate the relative effectiveness of Tetrapeptide-68 compared to other well-established anti-aging products.

Point 4: Why does tetrapeptide-68 possess anti-aging properties? How does it affect the skin? What is the mechanism behind the anti-aging property of tetrapeptide-68. This should be included in the discussion section in detail.

Point 4 response: We appreciate your insightful question regarding the anti-aging properties of Tetrapeptide-68. We have updated the discussion section of our manuscript to include the following information:

Tetrapeptide-68 has demonstrated significant anti-aging properties, primarily attributed to its elastase inhibitory activity and its effects on skin cell proliferation. Previous studies have confirmed that Tetrapeptide-68 effectively inhibits elastase, an enzyme responsible for the breakdown of elastin in the skin. Elastin is a critical protein that contributes to skin elasticity and firmness, and its degradation is a key factor in the formation of wrinkles and loss of skin structure.

In addition to its elastase inhibitory effects, Tetrapeptide-68 has been shown to promote skin cell proliferation. This enhancement in cell growth aids in the regeneration of skin tissues, contributing to the reduction of fine lines and wrinkles. (Line 312-319)

Our ongoing research aims to explore these mechanisms in depth to provide a more comprehensive understanding of how Tetrapeptide-68 exerts its anti-aging effects at the molecular level.

Point 5: If the subject is using other cosmetic products such as sun cream or serums, what are the probable side-effects of using test cream after other cosmetic products? Does the test cream have any interference with the activity of other cosmetic products?

Point 5 response: Thank you for raising this important concern regarding the potential side effects and interactions of the test cream when used in conjunction with other cosmetic products.

In our study, we have outlined specific criteria for participant discontinuation and removal from the study, which includes concerns related to the use of additional cosmetic products. The criteria are as follows:

Voluntary Withdrawal: Participants are allowed to withdraw from the study at any time. If a participant voluntarily withdraws or if there are personal reasons or objections to the clinical trial, they will be required to complete a 'Withdrawal of Consent Form' and will be excluded from further participation.

Adverse Reactions: If a participant experiences adverse reaction such as itching or redness at the test site, or any severe skin reactions that hinder the continuation of the study, they will be removed from the study and their data will be excluded.

Protocol Compliance: Participants must adhere to the study protocol, including the use of the test product as directed. The use of other products that could impact skin condition, such as medications or physical treatments, must be avoided. If participants use products that could affect the test results, such as other cosmetic products including sunscreens or serums, or fail to comply with the study protocol with adherence below 90%, they will be excluded from the study.

We have included detailed criteria for participant discontinuation and protocol compliance in the study protocol to address these concerns effectively. Further studies may be needed to explore the interactions between the test cream and various other cosmetic products in more detail.

Round 2

Reviewer 1 Report

Comments and Suggestions for Authors

The authors addressed all the concerns raised by this reviewer but one minor point: Although they described in their response that they added scales, there's no scale in Fig.4c. Please make sure to add a scale bar in the image and indicate the length of the scale in the figure legend.

Author Response

Point 1: The authors addressed all the concerns raised by this reviewer but one minor point: Although they described in their response that they added scales, there's no scale in Fig.4c. Please make sure to add a scale bar in the image and indicate the length of the scale in the figure legend.

Point 1 response: Thank you for your valuable feedback. I would like to address the point you raised regarding the absence of a scale bar in Fig. 4c. It is customary not to insert scale bars in images of wrinkles around the eyes. Instead, we have highlighted the improved areas using circles for clarity. We have updated the description of Figure 4c to indicate that the improved areas are marked with circles. (Line 252)

Reviewer 3 Report

Comments and Suggestions for Authors

All concerns have been addressed. The manuscript can be accepted with the current modifications. 

Author Response

Thank you for your thorough review and valuable feedback.
